# Luteolin alleviates CUMS-induced depressive-like behavioral deficits in mice through blocking the JAK2/STAT3 pathway

Hai-hong Wu[ID]1*, Shang-yue Chen2, Dong-liang Yuan3, Ting Zhao4

1 Pharmacy Intravenous Admixture Services, Baoji Central Hospital, Baoji, Shaanxi, China, 2 Pharmacy Department of Beijing Xiaotangshan Hospital, Beijing, China, 3 Department of Pharmacy, Tangdu Hospital, Air Force Military Medical University, Xi'an, China, 4 Department of Pharmaceutical Preparation, Baoji Central Hospital, Baoji, Shaanxi, China

* 604090113@qq.com

## Abstract

This study probed into the potential effects of luteolin (LUT) on depressive-like behavioral deficits caused by chronic unpredictable mild stress (CUMS) in mice, with a focus on its underlying molecular mechanisms. Western blot analysis revealed that CUMS notably activated the JAK2/STAT3 pathway, as indicated by elevated levels of phosphorylated JAK2 (p-JAK2) and p-STAT3. Treatment with LUT notably diminished p-JAK2 and p-STAT3, suggesting that LUT alleviates CUMS-induced depressive-like behavioral deficits by blocking the JAK2/STAT3 pathway. Behavioral assessments, including the forced swim and sucrose preference tests, demonstrated that LUT remarkably improved depressive-like symptoms. Furthermore, LUT treatment diminished the levels of pro-inflammatory cytokines, which were elevated by CUMS, further supporting the involvement of LUT in exerting antidepressant activities via the modulation of inflammatory responses. This study is the first to integrate multidimensional evidence from behavioral tests, neuroinflammation, and the JAK2/STAT3 signaling pathway, systematically demonstrating that luteolin alleviates CUMS-induced depression-anxiety comorbidity through synergistic regulation of an antioxidant–anti-inflammatory–neural signaling network.

Depression is a complex psychiatric disorder marked by chronic low mood, loss of interest in activities, and a range of cognitive and behavioral disruptions, all of which severely impact individuals' daily lives and social interactions [1]. Although selective serotonin reuptake inhibitors (SSRIs) are commonly prescribed to manage depression, challenges persist due to their slow onset, variable efficacy, and potential side effects. As a result, identifying alternative therapeutic strategies and novel drug targets has become a critical focus in depression research.

**Data availability statement:** All relevant data for this study are publicly available from the figshare repository (https://doi.org/10.6084/m9.figshare.30038803).

**Funding:** The author(s) received no specific funding for this work.

**Competing interests:** The authors have declared that no competing interests exist.

Recently, natural plant-derived bioactive compounds have garnered attention for their potential antidepressant effects and relatively low toxicity compared to conventional pharmaceuticals [2]. Among these, luteolin (LUT), a flavonoid naturally found compound, has demonstrated a variety of beneficial effects, including antioxidant, anti-inflammatory, and neuroprotective actions in various disease models [3]. Luteolin is primarily found in medicinal herbs such as honeysuckle, chrysanthemum, *Schizonepeta tenuifolia*, *Ajuga decumbens*, *Ligusticum chuanxiong*, *Salvia miltiorrhiza*, and safflower. Luteolin has previously been reported to exhibit activity against diabetes, diabetic complications, and Alzheimer's disease. Moreover, it contributes to the antidepressant-like effects, likely through potentiation of the GABAA receptor chloride ion channel [4]. Computational screening of phytochemicals derived from African medicinal plants also identified luteolin as a promising candidate, supported by a high docking score against hMAO-A with a binding energy of −98.5797 kcal/mol, indicating strong inhibitory potential [5]. In a comparative study of quercetin-related flavonoids and tea catechins on MAO-A activity in mouse brain mitochondria, luteolin demonstrated superior inhibitory efficacy compared to quercetin [6]. Further evidence confirms its potent inhibition of human MAO-A [7]. However, the precise molecular mechanisms by which LUT mediates its antidepressant effects remain poorly understood.

The pathogenesis of depression is closely linked to chronic stress, which can activate both the neuroendocrine and immune systems, potentially leading to brain dysfunction and behavioral abnormalities [8]. The JAK/STAT pathway is a key intracellular pathway involved in regulating gene expression, cell proliferation, and differentiation [9]. Emerging research suggests that this pathway is pivotal in the pathophysiology of depression, with abnormal activation of JAK2 and STAT3 potentially contributing to the onset and progression of depressive symptoms [10]. Therefore, examining the significance of LUT on the JAK/STAT pathway could offer valuable insights into its antidepressant mechanisms.

This study focuses on the impact of LUT on the JAK/STAT pathway in a mouse model of depression caused by chronic restraint stress. The primary objective is to establish a theoretical basis for the potential development of LUT as an antidepressant agent. By assessing the modulatory effects of LUT on depressive behavioral deficits, neurotransmitter levels, and the expression of key molecules in the JAK/STAT pathway, we aim to further elucidate its therapeutic potential and underlying mechanisms in the treatment of depression. This research could contribute to the development of novel therapeutic modalities for depression and enhance our understanding of the molecular basis of LUT's antidepressant effects.

## 1 Materials and methods

### 1.1 Animals

Due to the estrous cycle in female mice, which causes cyclical fluctuations in the levels of hormones such as estrogen, these fluctuations can significantly affect various physiological indicators including metabolism, immune response, neurological function, and drug response. Therefore, this study utilized Male C57BL/6

mice (aged 6–8 weeks, weight range: 18–25 g) sourced from the Experimental Animal Center of the Air Force Medical University for the experiments. Prior to the experiment, the mice were acclimatized for one week in a controlled environment, with a constant temperature of 24 ± 2°C, 50%–60% relative humidity, and a 12/12-hour light/dark cycle. Immediately following the behavioral tests, mice were anesthetized with 2% pentobarbital sodium (50 mg/kg) and brain tissue samples were collected from the lesion site. All procedures followed ethical guidelines and scientific standards (SCXK2019−001).

## 1.2 Drugs

LUT (CAS No.: 491-70-3) was procured from Taoshu Biotechnology Co (purity > 99%, Fig 1A). Ltd. For preparation, 10, 20, and 40 mg of LUT powder were weighed and dissolved in 2 mL of ethanol in 15 mL EP tubes. The resulting solution was then diluted with distilled water to a final volume of 10 mL.

## 1.3 Methods

**1.3.1 Model preparation.** After the acclimatization period, the mice were randomized into different groups based on body weight. The control (Con) group was kept under normal conditions (n = 8), while the CUMS and LUT treatment groups were subjected to CUMS stress. The detailed model preparation procedure is as follows:

Mice were individually restrained in homemade round restraint tubes for 3 hours each day, with random restraint times, over a period of 28 consecutive days. During the restraint period, food and water were withheld. The model group exhibited significant depression-like behavioral deficits, with the core evaluation criteria being: 1) a significantly reduced sucrose preference rate in the sucrose preference test (SPT), indicating the presence of anhedonia, a core symptom of depression; and 2) significantly prolonged immobility time in the forced swim test (FST) and tail suspension test (TST), suggesting a state of behavioral despair. Furthermore, the open field test (OFT) showed no significant difference in total distance traveled in the model group, ruling out the potential confounding effects of general locomotor activity on depression-like behavioral measures. Together, these behavioral results demonstrate the successful replication of the CUMS-induced depression model. The LUT treatment groups (LUT-L, LUT-M, and LUT-H) received LUT orally at doses of 10, 20, and 40 mg/kg, respectively (0.2 mL), 30 minutes prior to restraint for 3 consecutive weeks. The CUMS group received 0.5% DMSO, also 30 minutes before restraint, for 21 consecutive days.

**1.3.2 Open field test (OFT).** One day after the experiment, all animals were moved to the experimental room 30 minutes before testing. The environment was kept quiet with moderate lighting. Mice were then placed in a 60 × 60 cm open field arena and allowed to explore freely for 6 minutes. The amount of time spent in the central area (30 × 30 cm) and the number of entries into this central zone were recorded as measures of anxiety-likebehavioral deficits. To minimize

A B

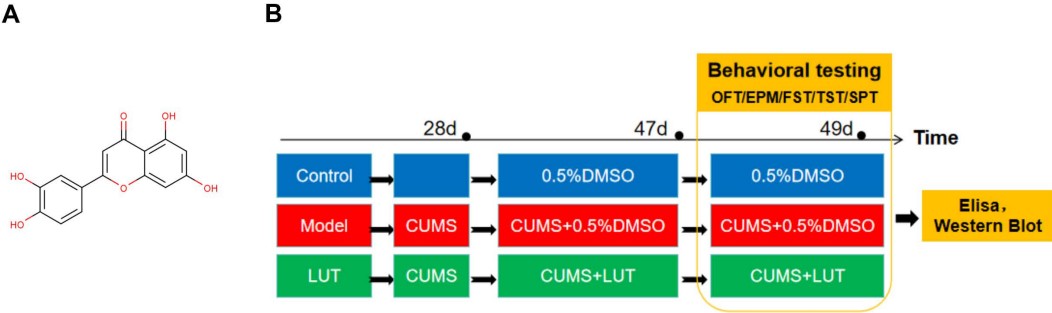

**Fig 1. (A) Luteolin structural formula. (B) Experimental Workflow of LUT on CUMS-induced Depressive-like Behavioral deficits in Mice.**

potential odor contamination between trials, the arena was thoroughly cleaned with alcohol before each session. Data analysis was made utilizing Tracking Master behavioral software.

**1.3.3 Elevated Plus Maze (EPM).** The EPM test is commonly adopted to assess anxiety-like behavioral deficits in rodents [11]. The maze consists of two open arms (5 × 50 cm) and two enclosed arms (5 × 50 cm) surrounded by 30 cm high walls. Mice were placed in the center of the maze and allowed to explore for 5 minutes. The time spent and the frequency of entries into the open arms were recorded as measures of anxiety-like behavioral deficits. Data analysis was made utilizing Tracking Master behavioral software.

**1.3.4 Forced Swimming Test (FST).** The FST, developed by Porsolt in 1977, is a well-utilized test for assessing depressive-like behavioral deficits in rodents, with high reliability and predictive validity [12]. Mice were placed into a transparent cylindrical container filled with water (22–25°C) to a depth of 15–20 cm. We considered the first 4 minutes as the acclimation phase and excluded this period from analysis. The immobility time of the mice during the subsequent 6-minute period was recorded for statistical evaluation. Following the test, mice were gently removed from the water, dried with towels, and returned to their cages. The Tracking Master behavioral software was selected for data analysis.

**1.3.5 Tail suspension test (TST).** The TST is another commonly applied behavioral assay to test depressive-like behavioral deficits in rodents by assessing immobility time while the animals are suspended. On the day of testing, each mouse was suspended by its tail from a hook about 50 cm above the ground and left undisturbed for 10 minutes (the first 4 minutes were considered the acclimation phase and were excluded from analysis). The duration of immobility was recorded as an indicator of depressive-like behavioral deficits. After the test, the mice were returned to their cages. Data were analyzed employing Tracking Master behavioral software.

**1.3.6 Sucrose preference test (SPT).** The SPT evaluates the function of the reward system and is often utilized to measure anhedonia, a key symptom of depression. Mice were first habituated to drinking sucrose water. During the adaptation phase, each cage was equipped with two bottles containing a 1% sucrose solution for a 24-hour period. Subsequently, one bottle held the 1% sucrose solution while the other contained plain water for another 24-hour phase. Moving on to the formal test, mice underwent a 24-hour period of food and water deprivation, following which they were presented with a choice between a bottle of 1% sucrose water and a bottle of plain water. The positions of the bottles were alternated every 6 hours. After 24 hours, the bottles were weighed, and the sucrose preference rate was determined: sucrose preference rate = (sucrose water consumption/ total liquid consumption) × 100%.

**1.3.7 Western blot (WB).** Immediately following the behavioral tests, hippocampal tissue samples were harvested, lysed, and then homogenized on ice. The protein concentration was determined utilizing a BCA protein assay kit. Equal amounts of protein (20 μg) were loaded and separated by SDS-PAGE before being transferred onto a 0.45 μm PVDF membrane. The membrane was blocked with 5% skim milk in TBST for 1 hour at room temperature and then incubated overnight at 4°C with anti-actin (1:10,000), anti-p-JAK2 (1:1,000), anti-p-STAT3 (1:1,000), anti-JAK2 (1:1,000), and anti-STAT3 (1:1,000). Protein bands were visualized employing the enhanced chemiluminescent solution, and detection was performed with the ChemiScope 6200 system. Quantitative analysis of the bands was carried out using ImageJ software.

**1.3.8 Enzyme-Linked Immunosorbent Assay (ELISA).** Blood samples were harvested via eye puncture following the behavioral tests. Mice were then euthanized, and their hippocampal tissue was immediately harvested. Serum levels of the inflammatory cytokines IL-1β, IL-6, and TNF-α were measured using respective ThermoFisher's mouse ELISA kits. In addition, the levels of IL-6 and TNF-α in the hippocampus were assessed utilizing the same ELISA kits.

**1.3.9 Statistical methods.** Each experiment was independently repeated a minimum of three times, yielding consistent outcomes. Statistical analyses were conducted utilizing GraphPad Prism software, version 9.0 (GraphPad Software, Inc., USA). All parameters were presented as mean ± SEM. Two-by-two group comparisons were analyzed utilizing two-way ANOVA followed by Tukey post hoc test or repeated-measures t-test with adjusted P value. Other results were analyzed utilizing a two-tailed paired t-test (for two groups) and one-way ANOVA followed by a Tukey post hoc test (for multiple groups). $P < 0.05$ or adjusted $P < 0.0083$ was considered as statistical significance.

## 2 Results

Successful CUMS model mice were randomized into the CUMS group and the LUT-L, LUT-M, and LUT-H groups (10, 20, and 40 mg/kg, respectively), with 8 mice per group. An additional 8 mice were assigned to the Con group. Both the Control (Con) group and the CUMS group received a daily intragastric (i.g.). injection of 0.5% DMSO (in saline) at a volume of 10 mL/kg body weight. The LUT treatment groups received LUT at doses of 10, 20, and 40 mg/kg once daily for 21 consecutive days. After treatment, behavioral assessments, including the OFT, EPM, FST, TST, and SPT, were implemented to evaluate the antidepressant effects of LUT on CUMS-induced depressive-like behavioral deficits. The experimental timeline is summarized in Fig 1B

### 2.1 LUT ameliorates spontaneous motor activity in CUMS mice

Motor abnormalities are commonly observed in depressed individuals and can significantly impact daily functioning. We then performed OFT to evaluate the effects of LUT on motor function in CUMS mice. As illustrated in Fig 2A, the total distance traveled by CUMS mice was notably reduced compared to the Con group, indicating impaired motor activity induced by CUMS stress. Following LUT treatment, particularly at the high dose (LUT-H), the total distance traveled significantly increased, approaching the levels observed in the Con group. This suggests that LUT effectively restores motor function in CUMS-induced anxiety-like mice (Fig 2B). Additionally, CUMS mice spent evidently less time in the central zone, which may indicate heightened anxiety-like behavior. However, in the LUT-H group, the time spent in the central zone was distinctly increased, suggesting that LUT may exert anxiolytic effects. (Fig 2C).

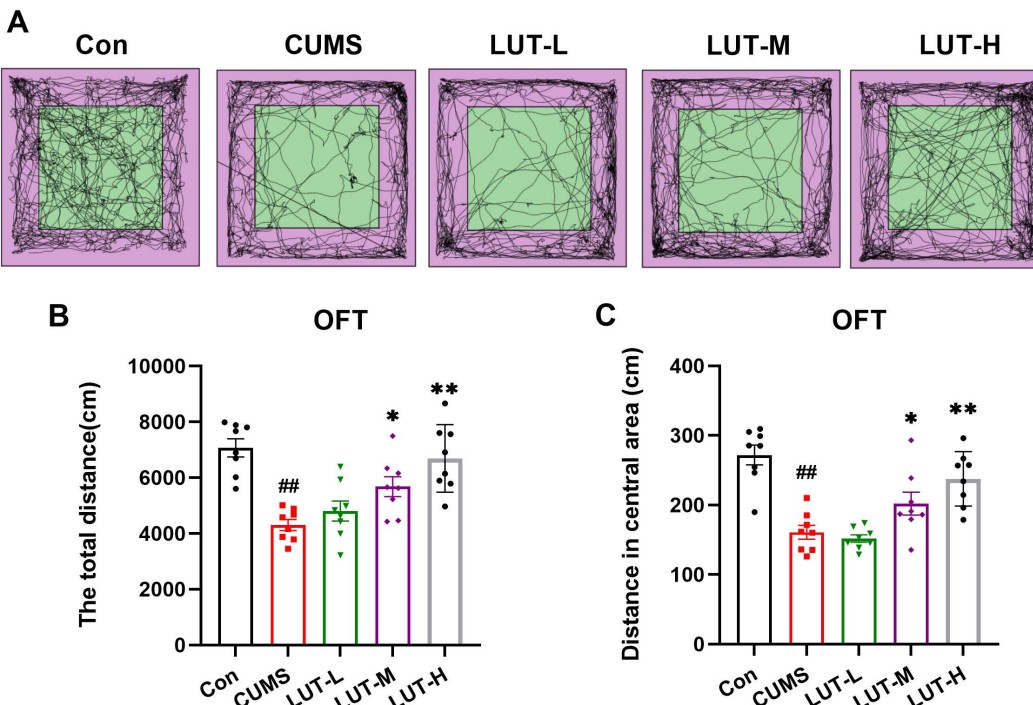

**Fig 2. LUT Ameliorates Abnormal Motor Function in CUMS Mice.** (A) Open Field Trajectory Map. (B) Total Distance Traveled in the Open Field. (C) Distance Traveled in the Center of the Open Field. Data are presented as mean ± SEM, with n = 8 per group. *P < 0.05, **P < 0.01 compared to the CUMS group. #P < 0.05, ##P < 0.01 compared to the Con group.

## 2.2 LUT mitigates anxiety-like behavior in CUMS mice

Anxiety-like behaviors are frequently observed in depressed individuals and can significantly impair daily functioning. To assess the effects of LUT on anxiety-like behavior in CUMS mice, an EPM test was performed. As demonstrated in Fig 3A, CUMS mice displayed a marked reduction in the number of entries into the open arms, which is indicative of anxiety-like behavior induced by CUMS stress. Following LUT (LUT-M, LUT-H) treatment, the number of entries into the open arms notably increased, suggesting that LUT effectively alleviates anxiety-like behavior in CUMS mice (Fig 3B, C). Although the other LUT doses showed a trend toward improvement, the differences were not statistically significant. Conclusively, these findings uncover that CUMS-induced anxiety-like behaviors are effectively reduced by LUT, especially at the high dose (LUT-H), indicating its potential anxiolytic properties in CUMS mice.

## 2.3 LUT ameliorates depression-like behavioral deficits in CUMS mice

The FST is a widely adopted method to assess despair-like behavior in rodents. The results revealed that the immobility time of CUMS mice in the FST was evidently longer than that of the Con group, indicating that CUMS stress induced depression-like behavioral deficits. In contrast, LUT-treated (LUT-M, LUT-H) mice exhibited a notable decline in immobility time relative to the CUMS group, with values approaching those of the Con group. Similarly, in the TST, CUMS mice spent significantly more time immobile than the Con group, but this was diminished following LUT (LUT-M, LUT-H) treatment. The SPT revealed that CUMS mice showed a decreased preference for sucrose, which is a hallmark of anhedonia. However, LUT treatment remarkably improved sucrose preference, suggesting a restoration of reward sensitivity. Collectively,

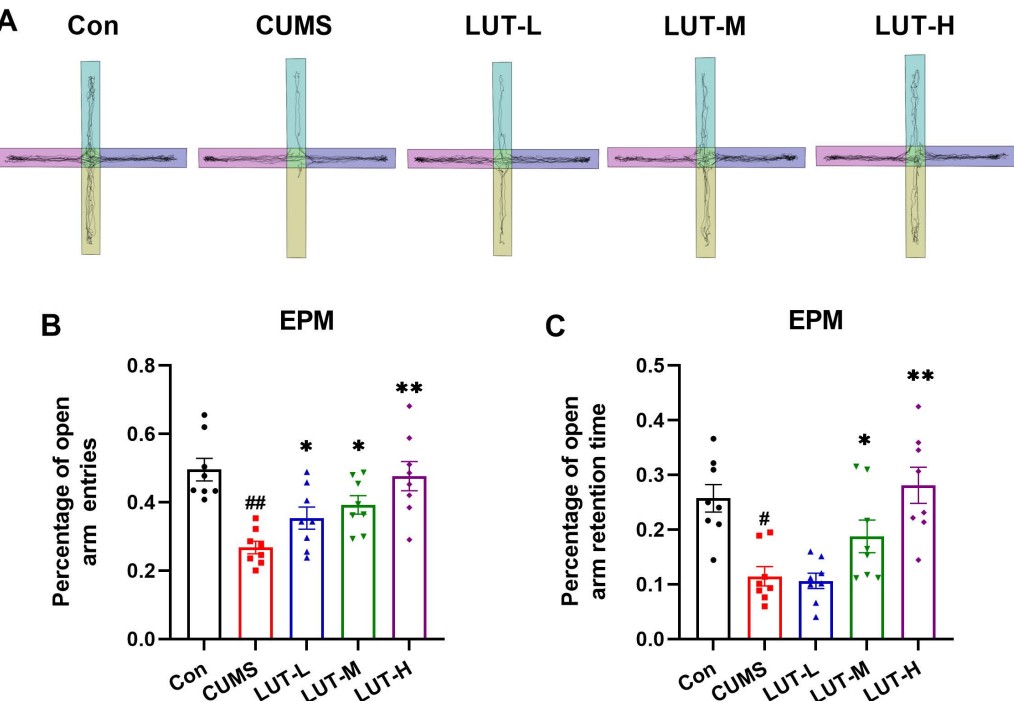

**Fig 3. LUT Attenuates Anxiety-like Behavior in CUMS Mice.** (A) Elevated Maze Trajectory Map. (B) Percentage of open arm entries in EPM. (C) Percentage of open arm retention time in EPM. Data are expressed as mean±SEM, with n=8 per group. *$P < 0.05$, **$P < 0.01$ compared to the CUMS group. #$P < 0.05$, ##$P < 0.01$ compared to the Con group.

these results confirm that LUT effectively alleviates various aspects of depression-like behavioral deficits in CUMS mice (Fig 4A–C).

## 2.4 LUT inhibits inflammatory cytokine levels in CUMS mice

ELISA analysis demonstrated that the levels of IL-1β, IL-6, and TNF-α in the plasma of CUMS mice were notably elevated relative to the Con group, highlighting a robust inflammatory response triggered by chronic unpredictable mild stress. However, following LUT (LUT-M, LUT-H) treatment, the levels of these inflammatory cytokines were evidently reduced, approaching those observed in the normal control group (Fig 5A–C). These findings confirm that LUT (LUT-M, LUT-H) may mitigate CUMS-induced depressive-like behavioral deficits in mice by inhibiting the associated inflammatory response.

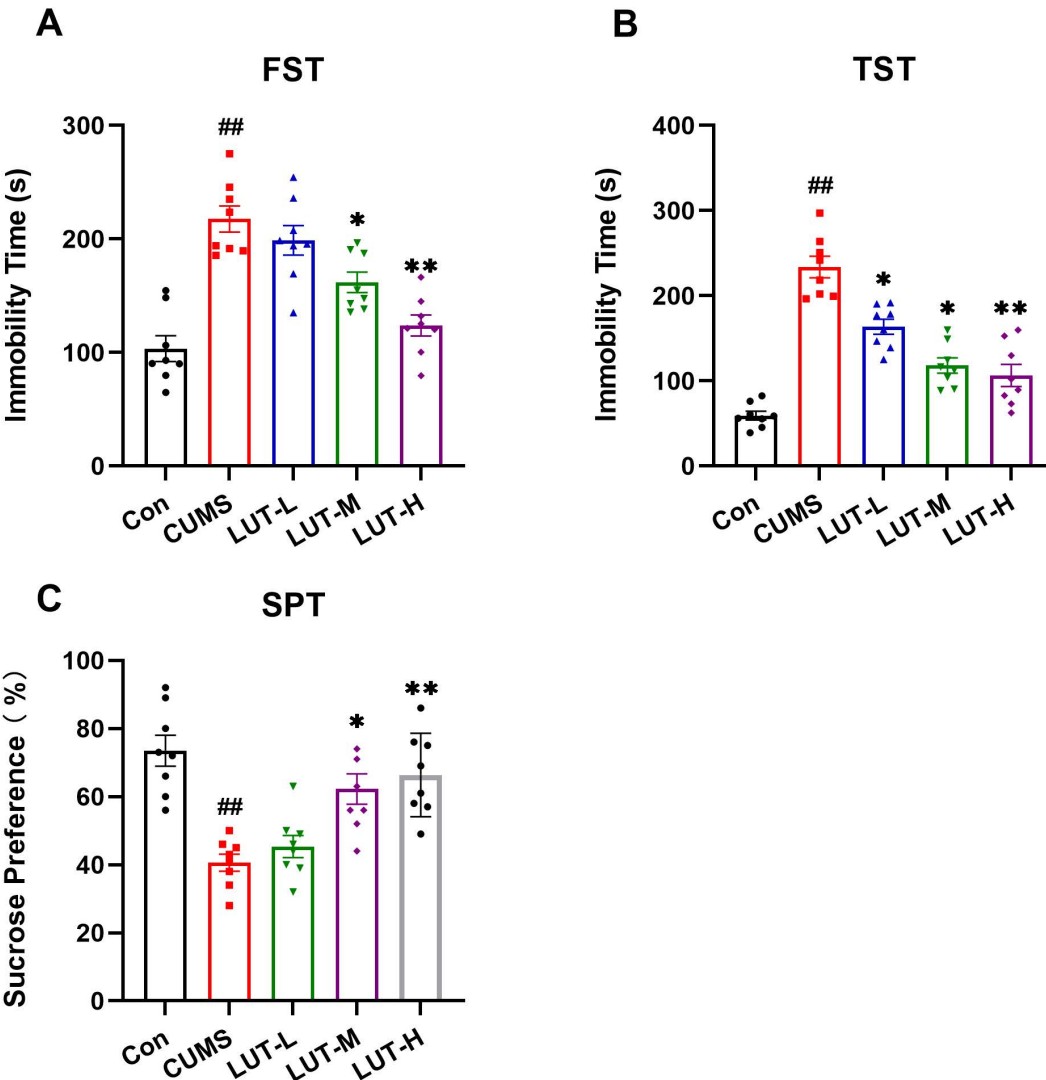

**Fig 4. LUT Attenuates Anxiety-like Behavior in CUMS Mice.** (A–C) FST, TST, and SPT results statistics. Data are expressed as mean±SEM, with n=8 per group. *$P<0.05$, **$P<0.01$ compared to the CUMS group. #$P<0.05$, ##$P<0.01$ compared to the Con group.

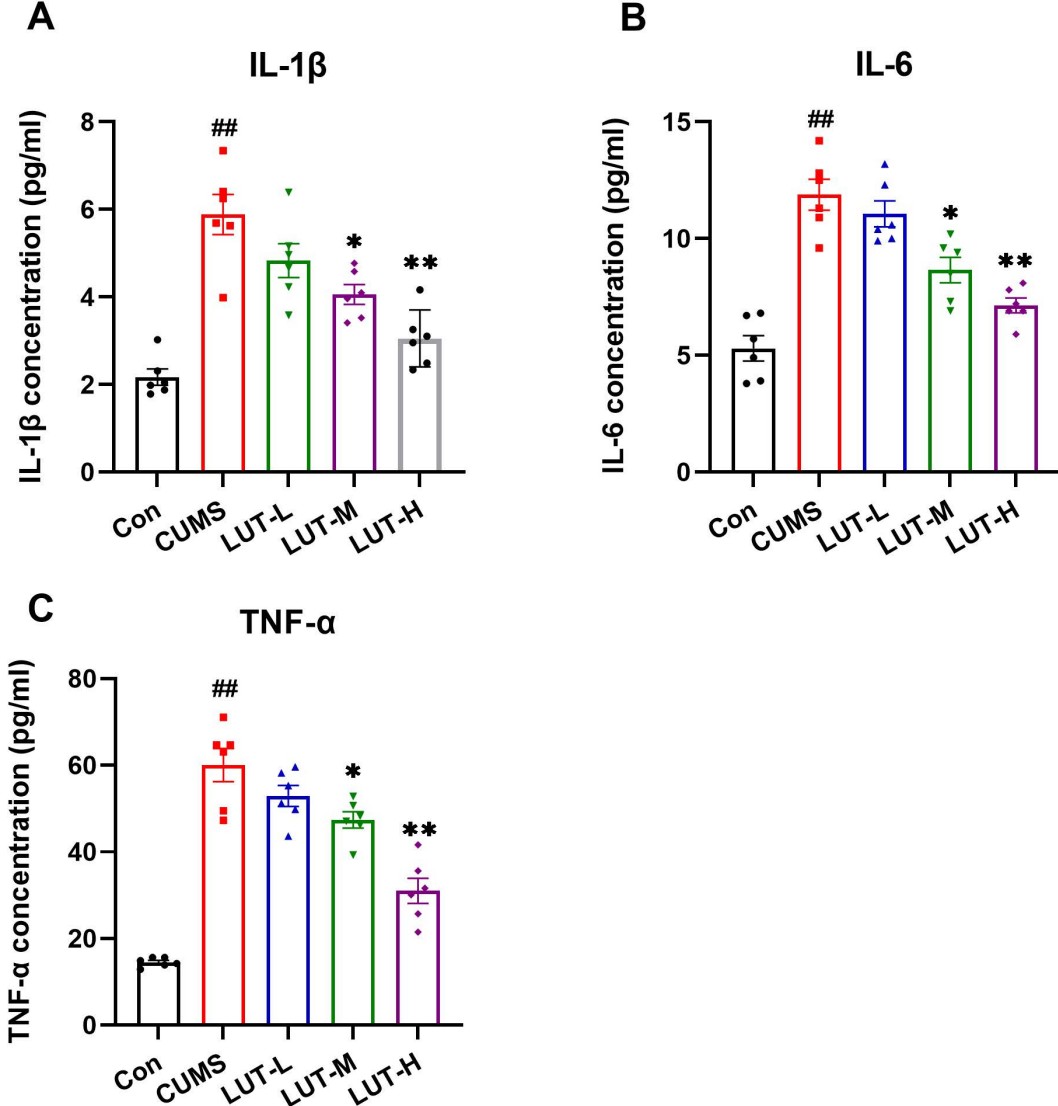

**Fig 5. LUT Inhibits Pro-inflammatory Cytokine Levels in Serum of CUMS Mice.** (A–C) LUT treatment significantly reduced the elevated levels of IL-1β (a), IL-6 (b), and TNF-α (c) in the serum, as shown by ELISA. Data are expressed as mean±SEM, with n=8 per group. *$P < 0.05$, **$P < 0.01$ compared to the CUMS group. #$P < 0.05$, ##$P < 0.01$ compared to the Con group.

## 2.5 LUT inhibits the JAK2/STAT3 pathway in CUMS mice

The JAK2/STAT3 signaling pathway is a critical pathway regulating cell proliferation, apoptosis, and inflammatory responses. Within this pathway, JAK2 is an upstream tyrosine kinase, and p-JAK2 represents its activated form. Activated p-JAK2 further phosphorylates and activates the downstream transcription factor STAT3. Phosphorylated STAT3 (p-STAT3) can translocate into the nucleus to initiate the transcription of target genes. To investigate whether luteolin exerts its effects through this pathway, we examined the expression of these key proteins. WB analysis of p-JAK2 and p-STAT3 demonstrated that relative to the Con group, the p-JAK2 and p-STAT3 levels were distinctly upregulated in CUMS mice. Following treatment with LUT (LUT-M, LUT-H) at varying concentrations, the levels of p-JAK2 and p-STAT3 were markedly decreased (Fig 6 A–C). These findings highlight that LUT (LUT-M, LUT-H) effectively alleviates

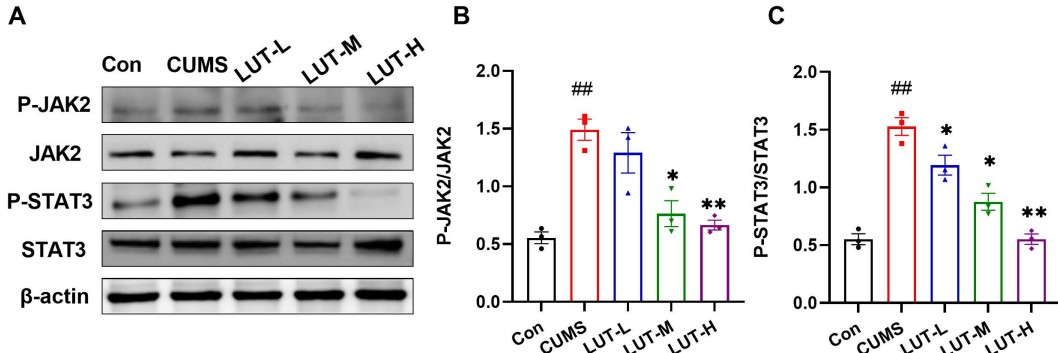

**Fig 6. LUT Inhibits the JAK2/STAT3 Signaling Pathway in CUMS Mice.** (A) Western blot analysis was used to detect the expression of p-STAT3, STAT3, p-JAK2, and JAK2 in the hippocampus. LUT treatment significantly reduced the levels of p-STAT3/STAT3 (B) and p-JAK2/JAK2 (C) in CUMS mice, with β-actin used as a loading control for normalization. Data are expressed as mean±SEM, with n=3 per group. **$P < 0.01$ compared to the CUMS group. #$P < 0.05$, ##$P < 0.01$ compared to the Con group.

CUMS-induced depressive-like behavior by blocking the JAK2/STAT3 pathway, confirming the critical role of this pathway in the pathogenesis of depression.

## Discussion

CUMS is a widely applied animal model for studying depression, as it simulates the prolonged stress that humans experience in daily life by exposing animals to multiple mild, unpredictable stressors over time [13,14]. This model induces a range of behavioral and biological changes similar to those detected in depressed patients, including anhedonia, sleep disturbances, appetite changes, and cognitive dysfunction [15,16]. The development of these symptoms in the CUMS model is linked to various neurobiological mechanisms, such as neuroinflammatory responses and dysregulation of key signaling pathways, including the JAK2/STAT3 pathway [16,17].

LUT, a flavonoid compound commonly found in plants, is known for its diverse biological characteristics, including antioxidant, anti-inflammatory, and neuroprotective capabilities [18–20]. Growing evidence suggests that LUT holds considerable therapeutic potential in the treatment of various neurological disorders, particularly in the prevention and management of depression [21,22]. Our study further corroborates these findings, demonstrating that LUT can evidently ameliorate anxiety and depressive-like behavioral deficits in mice subjected to CUMS. Our results validated that LUT effectively alleviated anxiety and depressive-like behavioral deficits in CUMS-induced mice. Behavioral assessments demonstrated that LUT treatment significantly ameliorated CUMS-induced behavioral deficits in mice. In terms of anxiety-like behaviors, the EPM and OFT indicated that LUT-treated mice exhibited significantly increased time spent in the open arms and greater movement distance in the central area, suggesting a notable anxiolytic effect. Regarding depressive-like behaviors, the FST and TSP showed a marked reduction in immobility time among LUT-treated mice, indicating decreased despair-like behavior. Meanwhile, the sucrose preference test revealed increased sucrose consumption, implying alleviation of anhedonia. Moreover, LUT treatment was shown to suppress the neuroinflammatory responses triggered by CUMS. This anti-inflammatory effect may be attributed to the compound's antioxidant and anti-inflammatory properties.

To evaluate the therapeutic effects of LUT on CUMS-induced depressive-like behavior, multiple behavioral tests were conducted, including the FST and the SPT. The results showed that CUMS mice exhibited notably increased immobility time in the FST and a marked decrease in sucrose preference in the SPT, both of which are typical indicators of depressive-like behavior. However, LUT treatment remarkably improved these behavioral outcomes, suggesting that LUT effectively alleviates CUMS-induced depressive-like behavior. These findings, together with alterations in signaling pathways and cytokine levels, further support the potential therapeutic value of LUT in the treatment of depression.

Additionally, we assessed the role of inflammatory factors in the CUMS model. The results revealed that CUMS elevated the levels of the three studied pro-inflammatory cytokines, which is consistent with previous studies that suggest CUMS exacerbates depressive-like behavioral deficits through the activation of inflammatory responses [21]. Interestingly, LUT treatment remarkably diminished the expression of these cytokines, indicating that LUT not only inhibits the activation of the JAK2/STAT3 pathway but also modulates inflammatory responses. This dual action may contribute to the alleviation of CUMS-induced anxiety and depressive-like behavioral deficits. The JAK2/STAT3 pathway is crucial in the pathophysiology of diverse psychiatric disorders, including depression [23]. Activation of JAK2 leads to the phosphorylation of STAT3, which then translocated to the nucleus to modulate the expression of genes engaged in inflammation, cell survival, and proliferation [24,25]. In the CUMS model, the activation of the JAK2/STAT3 pathway is thought to contribute to inflammatory responses and neuronal dysfunction, which in turn promotes depressive-like behavioral deficits [26,27]. Our study demonstrated that the overactivation of the JAK2/STAT3 pathway in CUMS mice was closely linked to neuroinflammation and the onset of depressive-like behavioral deficits. Specifically, the levels of p-JAK2 and p-STAT3 were notably elevated in the hippocampus of CUMS mice. Treatment with LUT effectively reduced the phosphorylation of these proteins, suggesting that LUT alleviates CUMS-induced depressive-like behavioral deficits by blocking the JAK2/STAT3 pathway. Moreover, considering LUT's impact on inflammatory cytokine levels, we propose that LUT exerts its antidepressant effects via a "dual-regulation" mechanism. This mechanism involves the concurrent inhibition of the JAK2/STAT3 pathway and the reduction of pro-inflammatory cytokines, which may together help alleviate depressive symptoms. This dual action provides strong theoretical support for the development of novel antidepressant therapies targeting the JAK2/STAT3 pathway.

In conclusion, our study reveals that LUT may offer a promising therapeutic strategy for depression through multi-pathway regulation. Specifically, LUT inhibits the JAK2/STAT3 pathway and reduces neuroinflammation, thereby improving depressive-like behavioral deficits. The main innovation of this study lies in revealing the multi-target mechanism of luteolin. For the first time, we have confirmed by correlating neuroinflammatory markers with pathway phosphorylation levels that luteolin inhibits p-JAK2/p-STAT3 activation, thereby reversing microglial over-activation and establishing an "antioxidant–anti-inflammatory–neural signaling regulation" pathway. These findings provide a novel mechanistic perspective and experimental evidence for the use of natural compounds in treating emotional comorbidities, highlighting their potential advantages of multi-target effects and low toxicity compared to existing synthetic drugs.

## Supporting information

**S1 File. Original images for blots and gels.**
(PDF)

## Author contributions

**Conceptualization:** Hai-hong Wu, Shang-yue Chen.

**Data curation:** Dong-liang Yuan.

**Investigation:** Ting Zhao.

**Methodology:** Hai-hong Wu, Dong-liang Yuan, Ting Zhao.

**Supervision:** Ting Zhao.

**Validation:** Dong-liang Yuan.

**Writing – original draft:** Hai-hong Wu.

**Writing – review & editing:** Hai-hong Wu, Shang-yue Chen.

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
