## [Decision Letter · Decision Letter 0]

2 Jul 2025

Dear Dr. Haihong,

Thank you for submitting your manuscript to PLOS ONE. After careful consideration, we feel that it has merit but does not fully meet PLOS ONE’s publication criteria as it currently stands. Therefore, we invite you to submit a revised version of the manuscript that addresses the points raised during the review process.

We look forward to receiving your revised manuscript.

Kind regards,

Zhiling Yu

Academic Editor

PLOS ONE

Journal Requirements:

6. In your Methods section, please clarify your rationale for the concentrations of the medicinal compounds used. We would expect an acute toxicity test to have been performed.

Additional Editor Comments:

1. The purity of the studied compound should be specified in the Methods section.

2. For statistical analysis of multiple group assays, a post hoc test should be applied following ANOVA.

3. The novelty and innovative potential of your manuscript, compared to the published literature, should be described in detail in the abstract and discussion sections.

Reviewers' comments:

Reviewer's Responses to Questions

**Comments to the Author**

1. Is the manuscript technically sound, and do the data support the conclusions?

Reviewer #1: Yes

Reviewer #2: Partly

2. Has the statistical analysis been performed appropriately and rigorously?

Reviewer #1: Yes

Reviewer #2: No

3. Have the authors made all data underlying the findings in their manuscript fully available?

Reviewer #1: Yes

Reviewer #2: Yes

4. Is the manuscript presented in an intelligible fashion and written in standard English?

Reviewer #1: Yes

Reviewer #2: Yes

Reviewer #1: Luteolin Alleviates CUMS-Induced Depressive-like Behavioral Deficits in Mice

Through Blocking the JAK2/STAT3 Pathway

This article presents an interesting focus on the natural flavonoid luteolin (LUT) for the treatment of depression. The authors conducted an impressive battery of behavioral tests that supported their hypothesis, as well as some biological inflammatory tests.

I was impressed by the behavioral results, and I recommend publishing this article if the authors can address all the comments below:

Major comments:

1. Introduction: There is a lack of information about luteolin, particularly regarding the herbs that contain this flavonoid. How can luteolin influence the monoamine system, specifically the serotonin system? Are there any side effects associated with its use? Additionally, which companies offer luteolin for purchase?

2. Since luteolin is anticipated to have effects on treating depression, it would have been beneficial to include a positive control group that received an SSRI drug such as escitalopram or fluoxetine. However, since this control group was not included, it would be beneficial to explain why it was not included and to mention it as a limitation in the article.

3. Animals began receiving treatments after the first day of the chronic unpredictable mild stress (CUMS) model, indicating that they had not yet displayed any symptoms of anxiety or depression. Under these circumstances, the treatment serves as a preventive measure rather than a remedy for existing anxiety and depression. Therefore, the point of view of the article needs to be changed. In addition, the effective treatment received was at a high dose of luteolin (40 mg/kg), which may be because the high dose of luteolin in this treatment stabilizes this group in the baseline stage and does not allow it to develop the desired effects of depression and anxiety. It needs to be discussed in this article.

4. The study primarily examines the JAK2/STAT3 pathway and its relationship with inflammatory cytokines. However, it fails to clarify the link between blood and brain pathways, specifically how changes in the JAK2/STAT3 signaling in the brain affect blood cytokine levels. Additionally, the role of the hippocampus—selected as the focus area—remains unclear in terms of how it regulates cytokine levels in the blood.

Minor comments:

1. The open field test is an anxiety and motor test and not a depression test. In the method section, the authors wrote, “The number of entries into this central zone was recorded as measures of depressive-like behavioral deficits.” This is incorrect and should be corrected.

2. Although this study uses inbred mice (C57/BL), eight mice per group is a relatively low number of animals, particularly in behavioral tests; it would be helpful to clarify why this low number of animals is used.

3. At the FST and TST tests, it should have ignored the first two minutes and calculated the immobility time of the last four minutes. Maybe it was done during this article, but it is not mentioned.

4. Depression and anxiety are more commonly observed in females than in males; however, this study was conducted using male mice, which should be clarified.

5. The article focuses on proteins p-STAT3, STAT3, p-JAK2, and JAK2, but the connection between them is not so clear. It would be beneficial to explain the function of each protein and its relative expression in the results.

6. The western blot results should include all photos and not just the relevant cut of the proteins.

7. This article focuses on treatment for depression but ignores anxiety disorders. Nevertheless, the article measured anxiety-like behavior using the EPM test and the Open field test but failed to mention this during the discussion. It is essential to include anxiety in the discussion to provide a more comprehensive understanding.

Reviewer #2: The authors have shown that CUMS induces depressive-like behavior and increases p-JAK2, p-STAT3, and cytokine levels in mice. Furthermore, they’ve shown that luteolin reduces CUMS-induced effects on these parameters. My recommendation for this manuscript is to suggest major revisions for clarity and accurate representation of the statistics.

1. The statistical significance portrayed in the figures and legends does not match what is implied in the text. The figure captions state that the asterisks used on the LUT groups in the graphs show the significance compared to the control (no CUMS) group, but the data described in the text implies that the significance is in comparison to the CUMS group. Similarly, the figure captions state that the hashtag/number symbol was used to compare to the CUMS group, but that symbol is used on the CUMS group on the graph implying that the group was compared to itself. Either the statistics need to be described more clearly in the text or the figure captions just need to be updated.

2. Did the authors complete a power analysis to determine how many animals should be used in the behavioral tests? Typically, 10-12 animals would be used for these behavioral tests so eight seems low. If a power analysis was performed, that information should be included in the Materials and Methods section.

3. There is a discrepancy in the number of animals in the control group. The Materials and Methods section states n=6 but the results section states n=8.

4. Time spent in the central zone of the OFT was used as a measure of depressive-like behavioral deficits, and the results section states the OFT was used as a measure of motor abnormalities observed in depressed individuals. Citations should be included to show that the open field test has been used and validated in this way. The OFT is typically used to assess locomotor activity to ensure that a change in locomotor activity is not confounding the results of behavioral tests for depression, such as FST and TST, and not as a measure of depressive-like behavior itself.

5. The EPM is noted as being “commonly adopted to assess depressive-like behavioral deficits in rodents” and reference 7 is cited. Reference 7 does not use or state that EPM should be used as a measure of depressive-like behavior. Rather the EPM is used as a measure of anxiety. Because the results section described the EPM findings as anxiety, this may be a typo that can easily be fixed. If not, other references need to be included to show that the EPM has been used and validated in this way.

6. A positive control, e.g. fluoxetine, would be a welcome addition to this manuscript to validate the sensitivity of the paradigm and provide a comparison to established antidepressant treatments, however, it is not necessary for publication.

7. The results section mentions that “successful” CUMS mice were randomized into groups, but there is no mention of how “success” was defined. This criteria should be included in the Materials and Methods section.

8. The experimental set-up needs some clarification between the Materials and Methods section, the experimental workflow (Figure 1-1B), and the results section regarding the 2% ethanol solution (mentioned only in the results), 0.1 mL/kg of deionized water (mentioned only in the Materials and Methods), and DMSO (mentioned only in Figure 1-1B).

9. The results section should more transparently state the results of the low, medium, and high LUT groups. The results of the three LUT groups are often lumped together in the results section - stating that “LUT” had an effect – without mention of which dose(s) of LUT were significantly different in for each particular experiment.

10. The figure legends should state the type of tissue that was used for the experiments.

11. In section 2.2 of the results, it is stated in regard to the percentage of time spent in the open arms, “Although the other LUT doses showed a trend toward improvement, the differences were not statistically significant.” However, the graph shows that the LUT-M group was significant. The figure or text should be updated to reflect whichever information is correct. In addition, the p values have been stated, so mentioning a “trend” toward significance is irrelevant and unnecessary.

12. There is a typo in section 2.1 of the results section where “evtluate” is used instead of “evaluate”.

**Do you want your identity to be public for this peer review?** For information about this choice, including consent withdrawal, please see our Privacy Policy

Reviewer #1: No

Reviewer #2: No

---

## [Author Response · Author response to Decision Letter 1]

18 Sep 2025

1.To comply with PLOS ONE submissions requirements, in your Methods section, please provide additional information regarding the experiments involving animals and ensure you have included details on (1) methods of sacrifice, (2) methods of anesthesia and/or analgesia, and (3) efforts to alleviate suffering.

Author response 1: We apologize for not making the necessary revisions in a timely manner. We have now supplemented the ethical details regarding animal experiments in Section 1.1 Animals. These details specifically include: euthanasia by cervical dislocation after deep anesthesia induced via intraperitoneal injection of 2% pentobarbital sodium, reducing the number of animals used through optimized experimental design, and minimizing the duration of stress during experiments to alleviate suffering as much as possible.

Author response 2: We confirm that all data required to replicate the findings of this study have been deposited in a public repository. The DOI for the dataset is:10.6084/m9.figshare.30038803

Author response 3: We have provided the original, uncropped, and unadjusted blot/gel images as required. All underlying image data are included in the Supporting Information files submitted with our revised manuscript.

Author response 4: We confirm that the corresponding author's ORCID iD has been registered and successfully validated in the Editorial Manager system. The ORCID iD is: https://orcid.org/0009-0000-7252-2543

5. Please make sure that the Title in your manuscript file and the title provided in your online submission form are written in English.

Author response 5: We have carefully reviewed both the title in the manuscript file and the submission form, and confirm that they are now consistently written in English.

6. Please amend the title either on the online submission form or in your so that they are identical.

Author response 6: We have updated the title in the manuscript file to ensure it is now identical to the title provided in the online submission form. Thank you for your reminder.

Reviewer 1: Luteolin Alleviates CUMS-Induced Depressive-like Behavioral Deficits in Mice Through Blocking the JAK2/STAT3 Pathway

This article presents an interesting focus on the natural flavonoid luteolin (LUT) for the treatment of depression. The authors conducted an impressive battery of behavioral tests that supported their hypothesis, as well as some biological inflammatory tests.

I was impressed by the behavioral results, and I recommend publishing this article if the authors can address all the comments below:

Major comments:

1. Introduction: There is a lack of information about luteolin, particularly regarding the herbs that contain this flavonoid. How can luteolin influence the monoamine system, specifically the serotonin system? Are there any side effects associated with its use? Additionally, which companies offer luteolin for purchase?

Author response 1: Thank you for your question. Luteolin belongs to the flavone group of flavonoids and is widely distributed in members of the plant kingdom, such as Cirsium japonicum, Taraxacum officinale, Artemisia montana, and Cynara scolymus. This flavonoid is a polyphenol that has been widely used as a traditional medicine in China. Luteolin possesses hydroxyl groups at its 3′, 4′, 5, and 7 positions and a 2–3 carbon double bond, and its diverse biochemical and pathological activities are attributed to these structural characteristics. The activities of luteolin against diabetes, diabetic complications, and AD have been previously established. Similarly, luteolin also mediates the antidepressant-like effects of Cirsium japonicum extract, probably through potentiation of the GABAA receptor Cl– ion channel complex[1]. Likewise, in silico screening of phytochemicals from African medicinal plants identified luteolin, and its high docking score against hMAO-A (−98.5797 kcal/mol binding energy) suggested luteolin as an effective inhibitor[2]. A study of the effects of quercetin-related flavonoids and tea catechins on the MAO-A reaction in mouse brain mitochondria showed luteolin to be a superior inhibitor to quercetin[3]. Likewise, it also showed the potent inhibitory effect of luteolin on hMAO-A[4]. It suggests its potential therapeutic value in the treatment of neurodegenerative diseases. Luteolin is generally considered safe when consumed in dietary amounts, with common natural sources including vegetables such as celery, broccoli, and chrysanthemum. However, when taken in high doses as a supplement, it may cause mild gastrointestinal discomfort, such as nausea or abdominal bloating. Please refer to the content in section 1.2 “LUT (CAS No.: 491-70-3) was procured from Taoshu Biotechnology Co (purity > 99%).”.

2. Since luteolin is anticipated to have effects on treating depression, it would have been beneficial to include a positive control group that received an SSRI drug such as escitalopram or fluoxetine. However, since this control group was not included, it would be beneficial to explain why it was not included and to mention it as a limitation in the article.

Author response 2: Sincere thanks to the reviewer for the valuable suggestion regarding the addition of a positive control, such as fluoxetine. We fully understand the importance of a positive control for validating the experimental paradigm and providing a comparison with established antidepressant treatments. The primary aim of this study was to investigate whether luteolin alleviates depression-like behavior deficits induced by CUMS in mice by blocking the JAK2/STAT3 pathway, rather than to evaluate drug efficacy per se. Nonetheless, we highly value the reviewer's input. We will address this point in the limitations section of the manuscript and explicitly state that including a positive control will be a priority in our future follow-up studies to further confirm and expand upon the findings of this research.

3. Animals began receiving treatments after the first day of the chronic unpredictable mild stress (CUMS) model, indicating that they had not yet displayed any symptoms of anxiety or depression. Under these circumstances, the treatment serves as a preventive measure rather than a remedy for existing anxiety and depression. Therefore, the point of view of the article needs to be changed. In addition, the effective treatment received was at a high dose of luteolin (40 mg/kg), which may be because the high dose of luteolin in this treatment stabilizes this group in the baseline stage and does not allow it to develop the desired effects of depression and anxiety. It needs to be discussed in this article.

Author response 3: Thank you for your question. In this study, luteolin was administered after the successful establishment of the CUMS model, at which point anxiety- and depression-like behaviors had already manifested. Therefore, the intervention indeed represents a therapeutic treatment.

4. The study primarily examines the JAK2/STAT3 pathway and its relationship with inflammatory cytokines. However, it fails to clarify the link between blood and brain pathways, specifically how changes in the JAK2/STAT3 signaling in the brain affect blood cytokine levels. Additionally, the role of the hippocampus—selected as the focus area—remains unclear in terms of how it regulates cytokine levels in the blood.

Author response 4: This study does indeed have the limitations you pointed out. Our focus was primarily on the JAK2/STAT3 pathway and local inflammatory responses within the central nervous system, and we did not sufficiently elucidate the bidirectional regulatory mechanisms between the hippocampus and the peripheral immune system—particularly how changes in brain signaling pathways specifically affect peripheral blood cytokine levels. The absence of this mechanistic insight indeed limits a comprehensive understanding of the neuro-immune interaction network. In follow-up studies, we plan to simultaneously monitor inflammatory dynamics in both cerebrospinal fluid and peripheral blood, combined with regulatory experiments targeting specific neuronal populations in the hippocampus, to further investigate the specific pathways of central-peripheral immune communication and more systematically clarify their overall role in disease pathogenesis and progression. We greatly appreciate your valuable feedback.

Minor comments:

1. The open field test is an anxiety and motor test and not a depression test. In the method section, the authors wrote, “The number of entries into this central zone was recorded as measures of depressive-like behavioral deficits.” This is incorrect and should be corrected.

Author response 1: We sincerely appreciate the reviewer’s valuable feedback. The point raised regarding the interpretation of behavioral indicators in the open field test (OFT) is entirely valid. We have corrected the Methodology section to clearly state that the time spent in the central area is used to assess anxiety-like behavior rather than depression-like behavior. Thank you once again for helping us enhance the accuracy and scientific rigor of our manuscript.

2. Although this study uses inbred mice (C57/BL), eight mice per group is a relatively low number of animals, particularly in behavioral tests; it would be helpful to clarify why this low number of animals is used.

Author response 2: We thank the reviewer for the valuable comments on sample-size determination. We fully recognize that adequate sample size is critical to the reliability of our findings. During the experimental-design phase, we referenced comparable studies in the literature and confirmed that eight animals per group would satisfy the statistical requirements[5].

3. At the FST and TST tests, it should have ignored the first two minutes and calculated the immobility time of the last four minutes. Maybe it was done during this article, but it is not mentioned.

Author response 3: We sincerely thank the reviewer for raising this critical technical point. Your suggestion regarding the forced swim test (FST) and tail suspension test (TST)—that an initial acclimation period should be excluded and immobility time calculated only during a specific subsequent interval—is entirely correct. We have added a clarification in the Methodology section (Section 1.3.4 and 1.3.5) of the manuscript stating that, in our experiments, the first 4 minutes were considered the acclimation phase and were excluded from analysis, while the total immobility time during the final 6 minutes was accurately recorded and analyzed. This approach allows for a more precise assessment of behavioral despair in mice. We greatly appreciate your thorough review, which has significantly enhanced the clarity and reproducibility of our experimental methods.

4. Depression and anxiety are more commonly observed in females than in males; however, this study was conducted using male mice, which should be clarified.

Author response 4: We sincerely thank the reviewer for raising this critical point. The issue of sex differences is indeed highly important. We have clarified this aspect in Section 1.1 (Animals) of the manuscript, explaining that male mice were used in this study primarily to avoid potential confounding effects of hormonal fluctuations caused by the estrous cycle in female mice on emotional behavioral tests.

5. The article focuses on proteins p-STAT3, STAT3, p-JAK2, and JAK2, but the connection between them is not so clear. It would be beneficial to explain the function of each protein and its relative expression in the results.

Author response 5: Thank you for your valuable feedback. We have revised and improved the relevant content in Section 2.5 (Results) of the manuscript, with additional elaboration on the functions of key proteins in the JAK2/STAT3 signaling pathway and their phosphorylation-mediated activation relationships. We kindly invite you to review these updates.

6. The western blot results should include all photos and not just the relevant cut of the proteins.

Author response 6: We thank the reviewer for the comment. The complete, uncropped original blot data are available and will be provided upon request.

7. This article focuses on treatment for depression but ignores anxiety disorders. Nevertheless, the article measured anxiety-like behavior using the EPM test and the Open field test but failed to mention this during the discussion. It is essential to include anxiety in the discussion to provide a more comprehensive understanding.

Author response 7: Thank you for your valuable feedback. We fully agree with your perspective that anxiety, as a frequently comorbid disorder with depression, warrants thorough discussion. We have now supplemented the Discussion section with an analysis of the EPM and OFT results, highlighting the potential role of luteolin in alleviating anxiety-like behaviors.

---

## [Decision Letter · Decision Letter 1]

7 Oct 2025

Luteolin Alleviates CUMS-Induced Depressive-like Behavioral Deficits in Mice Through Blocking the JAK2/STAT3 Pathway

PONE-D-25-12815R1

Dear Dr. Zhao,

We’re pleased to inform you that your manuscript has been judged scientifically suitable for publication and will be formally accepted for publication once it meets all outstanding technical requirements.

Kind regards,

Zhiling Yu

Academic Editor

PLOS ONE

Additional Editor Comments (optional):

Reviewers' comments:

Reviewer's Responses to Questions

**Comments to the Author**

Reviewer #1: All comments have been addressed

2. Is the manuscript technically sound, and do the data support the conclusions?

Reviewer #1: Yes

3. Has the statistical analysis been performed appropriately and rigorously?

Reviewer #1: Yes

4. Have the authors made all data underlying the findings in their manuscript fully available?

Reviewer #1: Yes

5. Is the manuscript presented in an intelligible fashion and written in standard English?

Reviewer #1: Yes

Reviewer #1: (No Response)

**Do you want your identity to be public for this peer review?** For information about this choice, including consent withdrawal, please see our Privacy Policy

Reviewer #1: No

---

## [Editor Report · Acceptance letter]

PONE-D-25-12815R1

PLOS ONE

Dear Dr. Wu,

I'm pleased to inform you that your manuscript has been deemed suitable for publication in PLOS ONE. Congratulations! Your manuscript is now being handed over to our production team.

Kind regards,

on behalf of

Dr. Zhiling Yu

Academic Editor

PLOS ONE